# Application of Phyto-Stimulants for Growth, Survival Rate, and Meat Quality Improvement of Tiger Shrimp (*Penaeus monodon*) Maintained in a Traditional Pond

**DOI:** 10.3390/pathogens11111243

**Published:** 2022-10-27

**Authors:** Esti Handayani Hardi, Rudi Agung Nugroho, Maulina Agriandini, Muhammad Rizki, Muhammad Eko Nur Falah, Ismail Fahmy Almadi, Haris Retno Susmiyati, Rita Diana, Nurul Puspita Palupi, Gina Saptiani, Agustina Agustina, Andi Noor Asikin, Komsanah Sukarti

**Affiliations:** 1Department of Aquaculture, Faculty of Fisheries and Marine Science, University of Mulawarman, Samarinda 75242, Kalimantan Timur, Indonesia; 2Department of Biology Science, Faculty of Science Mulawarman University, Samarinda 75123, Kalimantan Timur, Indonesia; 3Study Program of Aquaculture, Polytechnic of Lingga, Lingga 29872, Kepulauan Riau, Indonesia; 4Faculty of Law, University of Mulawarman, Samarinda 75119, Kalimantan Timur, Indonesia; 5Faculty of Forestry, University of Mulawarman, Samarinda 75116, Kalimantan Timur, Indonesia; 6Faculty of Agriculture, University of Mulawarman, Samarinda 75117, Kalimantan Timur, Indonesia; 7Departement of Fisheries Product Technology, Faculty of Fisheries and Marine Science, University of Mulawarman, Samarinda 75242, Kalimantan Timur, Indonesia

**Keywords:** amino acids, fatty acids, natural fish drug, plant extract, silvofishery, tiger shrimp

## Abstract

The tiger shrimp culture in East Borneo is commonly performed using traditional pond system management. In this work, the objective was to evaluate the application of *Boesenbergia pandurata* and *Solanum ferox* extract supplemented as feed additives considering shrimp growth, survival rate, and meat quality culture in a traditional pond. There were three dietary groups that were stocked with 300 shrimp in this study. The shrimp were maintained in a pond, separated with a 3 × 3 m^2^ net. The dietary treatment applied was divided into three types, namely P1, without the extracts; P2, 20 mL kg^−1^ dietary supplementation; and P3, 30 mL kg^−1^ dietary supplementation in the diet. The findings revealed that the herb extract influenced the growth rate, feed efficiency, survival rate, and meat quality of the shrimp, mainly the amino and fatty acid contents in the shrimp meat. The 30 mL kg^−1^ herb extract dose in group 3 showed a higher growth performance and survival rate. In group 3, 98% of the shrimp could survive until the final study period, while 96% of shrimp survived in group 2, and 70% of the shrimp survived in group 1. These findings indicate that the phytoimmune (*B. pandurata* and *S. ferox)* extract can be utilized as a feed additive to improve the growth, survival rate, and meat quality of the shrimp.

## 1. Introduction

Tiger shrimp (*Penaeus monodon*) has become an excellent aquaculture commodity in East Borneo. The data show that tiger shrimp comprises the world’s second largest fishery sector [1,2]. In East Borneo, the production value of tiger shrimp reached 27,506 tons in 2020 [3]. Therefore, tiger shrimp culture is a sector that could increase the regional income in East Borneo, besides that from coal. Shrimp cultured with the silvofishery system has been applied in some countries, namely Vietnam, Taiwan, and Thailand [4]. Silvofishery is a cultivation system for brackish water that has a low input, sustainable aquaculture, and is integrated with mangrove [5]. The traditional shrimp ponds in Indonesia cover a larger area than intensive ponds, but only about 20% of the land is used for active aquaculture. It has numerous issues, including water quality supply, SPF larva, and lack of farmer technology knowledge. It is believed that by only using intensification, greater production can be obtained, and amaranth management has no drawbacks for improving production performance, whereas most communities manage their ponds in an extensive system with 36 ha per plot [6]. Pond management still relies on tides for the culture production cycle, as well as depending on live feed availability without water management handling either before or after the culture production process [7]. The occurrence of harvest failures is commonly caused by disease incidence [8]; high water quality fluctuation, mainly in DO and pH parameters; a slow growth rate; and high mortality level due to less available feed. Based on these problems, the utilization of plant extracts can achieve meaningful application, as plant extracts can help improve the growth performance and immunity level, besides increasing the amino and fatty acid quality in the shrimp meat, as well as being an additional nutrition source in shrimp feed [9,10,11,12,13,14].

The Indonesian Government, through the Ministry of Marine and Fisheries, has established a regulation regarding plant extract development as a standardized natural fish drug, as well as antibiotic and growth promoter usage limitation and restriction actions for cultured animals [15,16]. As a further action to prove the efficacy of plant extracts in shrimp culture, it is necessary to conduct a field experiment of plant extract formulations that have passed limited laboratory tests on a wider culture scale. The natural fish drug candidates made from ginger (*Boesenbergia pandurata*) and hairy-fruited eggplant (*Solanum ferox*) extracts have several phytoimmune compounds, namely flavonoids, alkaloids, and steroids, which function as antibacterial agents [7,17,18,19], immunostimulants [20], growth promoters [21], survival rate improvement agents [22], and additive ingredients in order to improve feed efficiency and growth performance [23]. This article will explain how these phyto-stimulants from *S. ferox* and *B. pandurata* extracts improvement the growth performance, immunomodulatory effect, mortality suppression, and meat quality of tiger shrimp maintained in the pond.

## 2. Materials and Methods

### 2.1. Culture Location

The pond used for the shrimp culture was a 1 Ha silvofishery pond in Solo Palai Village, Muara Badak Subdistrict, Kutai Kartanegara District, East Borneo (0°21′54.702″ S and 117°26′45.5028″ E). The pond was formed as a ditch, surrounding a *Rhizophora apiculata* mangrove plant with 1 m spacing and 80% density. Pond management was carried out in a traditional way, whereas the water exchanges followed the tidal cycle with a stocking density of 20 shrimp m^−3^.

### 2.2. Experimental Animal

The experimental activity was performed for 40 days in October 2021. The tiger shrimp seeds used originated from the Center for Brackish Water and Marine Hatchery, Manggar, Balikpapan, East Borneo. Before stocking the pond, shrimp seeds with an average weight of 0.01 g were first maintained in the hatchery in order to reach an average weight of 2.26 ± 0.04 g. 

The shrimp were previously selected based on their organ completion, marked by a transparent body color and incurved tail, less pigmented spots, well-formed eyes and eye-stalks, active-swimmers with a straight body and highly responsive to shock cues, reddish or pale color absence, and no glows observed in a dark room condition.

The maintenance tank in the hatchery was a 1 × 3 m squared tank filled with water at 0.75 m. The tiger shrimp maintenance tank in the pond was a 1 × 1 m net. As each treatment contained four replications, 12 nets were used to stock 100 shrimp seeds per net (n = 100 × 3 × 4 = 1200).

### 2.3. Plant Extracts

The plant extract used in this study was derived from *Boesenbergia pandurata* and *Solanum ferox.* It was obtained from Sempayau Village, Kutai Timur District, East Borneo, and the extraction process was performed using 98% ethanol [10]. The ingredient concentrations used comprised *B. pandurata* at 900 mg L^−1^ and *S. ferox* at 400 mg L^−1^ with a ratio of 2:1. The diets were produced by mixing the plant extract in the formulation following the diet ingredient composition in Table 1.

The diet ingredients are described in Table 1. The ingredients were mixed with warm water (38 °C) until forming a homogenous dough, before being supplemented with the phytoimmune extract based on the applied doses. Each diet was extruded with a mini-pelletizer and dried under sunlight. The dried pellets from different diet types were packed separately in airtight plastic containers and preserved in a refrigerator for further use. 

The shrimp were maintained for 40 days and fed with the formulated diets at 5% of the shrimp weight. Feeding was performed three times a day. The diet treatment groups in this study were divided into the following groups:

P0 = tiger shrimp fed without dietary extract supplementation

P1 = tiger shrimp fed with 20 mL kg^−1^ phytoimmune-supplemented diet

P2 = tiger shrimp fed with 30 mL kg^−1^ phytoimmune-supplemented diet

### 2.4. The Mortality and Growth

The mortality and growth parameters were measured every 10 days during the maintenance period, containing the total living shrimp, average body weight (ABW), average daily growth (ADG), and specific growth rate (SGR), based on the Aftabuddin et al. (2017) method [1]. The immunostimulatory activity of the shrimp was also observed at the final maintenance period, containing the total hemocytes (TH), phenol oxidase activity (PO), and anion-superoxide concentration (SO). 

### 2.5. The Hemolymph

The hemolymph sample was taken at 100 mL with a 1 mL syringe filled with 0.9 mL anticoagulant (trisodium citrate 30 mM, NaCl 115 mM, and EDTA 10 mM, pH 6–7). The total hemocytes (TH) were counted using a Neubaeur hemocytometer [24]. The hemolymph–anticoagulant mixture (100 mL) was dropped in a hemocytometer and the hemocytes were counted in four different squares under the microscope; each point was presented as cell mL 1 hemolymph. 

### 2.6. The Phenol Oxidase Activity (PO)

The PO of the hemocytes was determined using a spectrophotometer and L-dihydroxyphenylalanine (L-DOPA) as a standard [25]. Then, 50 mL of the hemolymph–anticoagulant mixture was mixed with 50 mL of SDS 10% and 1.0 mL of L-dihydroxyphenylalanine (0.19% L-DOPA in Tris-HCl buffer), and then incubated for 30 min at 25 °C on 96-microliter plates. The dopachrome formation was measured every 30 s for 3 min in a spectrophotometer at a 490 nm wavelength. The PO activity was presented as the dopachrome formation per 50 mL of hemolymph. 

### 2.7. The Amino and Fatty Acids

The meat quality of the shrimp (amino and fatty acids) was determined at the final maintenance period, while the water quality of the shrimp culture was measured once every 4 days, including temperature (27 ± 2 °C), salinity (16 ± 2 ppt), pH (7.6 ± 0.3), and DO (3–5 ppm). The measurement was performed in situ using a Waterproof Meter-HI98196 tool once every 4 days, in the morning and afternoon.

### 2.8. Data Analysis

The statistical analysis was carried out using STATISTICA v 13.2 (Statsoft Inc., Tulsa, OK, USA). For growth performance and survival rate parameters, different diet treatment groups were compared with each other using two-way ANOVA, followed by Tukey’s test. The alpha was determined at 0.05 for all of the analyses.

## 3. Results

### 3.1. The Mortality and Growth

The mortality level (%) of the tiger shrimp maintained in the pond was low, at 10-20% for the plant extract-supplemented diet treatment groups (Figure 1), but reached 34% for the control diet treatment group. These data represent low total mortality levels of the tiger shrimp culture in the pond. A significant difference (*p* < 0.05) between the plant extracts supplemented diet and the control diet treatment groups indicates that the plant extract of *B. pandurata* and *S. ferox* could help improve the body condition sustainably during the ongoing culture activity.

The shrimp growth performance (Table 2) was maintained for 40 days and those fed the 20-30 mL kg^−1^ plant extract supplemented diets weighed more than those without the plant extracts dietary supplementation, i.e., the average body weight, ABW; average daily growth, ADG; and specific growth rate, SGR. These growth performance parameters were significantly different between the plant extracts supplemented diet and control diet treatment groups (*p* < 0.05). The 20 and 30 mL kg^−1^ diet dose had no significant difference (*p* > 0.05) on the ADG and SGR values, but showed a significant difference for the ABW value (*p* < 0.05).

### 3.2. Immunomodulatory Activity

The total hemocytes of *P. monodon* after being fed with the phytoimmune-supplemented and controls diet are presented in Table 3. A significant difference (*p* < 0.05) was revealed among the different doses on the 10th day. A significant difference with the control diet treatment group was presented on the 20-th day in the 30 mL kg^−1^ diet dose, then the total hemocytes were found to be significantly different among the treatment groups on the 30th and 40th days.

The maximum phenol oxidase activity (PO) was found in the 30 mL kg^−1^ phytoimmune extract dose treatment group from the 10th to the 40th day of the observation period (Table 3). The anion superoxide concentration (SO) gradually increased in all treatment groups during the maintenance period (Table 3). The SO formation in the 20 and 30 mL kg^−1^ dose treatment groups was significantly higher on the 30th day of the maintenance period compared with in the control treatment group (*p* < 0.05).

### 3.3. Amino and Fatty Contents of Shrimp

The culture product quality, for example with shrimp, is mainly influenced by the culture process. The traditional maintenance system produces a more excellent product than the intensive system. The application of phytoimmune extracts made from *B. pandurata* and *S. ferox* can help improve several shrimp amino acids (body and head) (Table 4) and fatty acids (Table 5).

Table 4 indicates that the amino acids in both the shrimp body and head fed with 30 mL kg-1 of a phytoimmune-extract-supplemented diet treatment were higher than for the control diet treatment, although there was a small difference in value for the L-Alanine content in shrimp meat for the P0 treatment as the control diet treatment at 34,077.31 mg kg^−1^ compared with the P2 treatment as the plant extract-supplemented diet treatment at 34,275.23 mg kg^−1^. The attractive condition occurred in the L-Arginine and Taurine contents of the shrimp head, which were higher than the shrimp body meat.

Almost all of the fatty acids detected in the shrimp meat and head had a higher value in the 30 mL kg-1 phytoimmune-extract-dose treatment than in the control treatment. Several fatty acids were measured, but they remained undetected or showed a below threshold value, namely C 24:1 W9 (Nervonic Acid); C 18:1 W9T (T-Oleic Acid); C 23:0 (trichosanoic acid); C 8:0 (caprylic acid); C 15:1 (pentadecenoic acid); C 20:3 W6 (eicosatrienoic acid/W6); C 14:1 (miristoleic acid); C 13:0 (tridecanoic acid); C 18:3 W6 (linolenic acid/W6); C 11:0 (undecanoic acid); C 4:0 (butyric acid); C 18:2 W6T (T-linoleic acid); C 6:0 (caporic acid); C 20:3 W3 (eicosatrienoic acid/W3); C 22:1 (erucic acid); C 22:0 (behenic acid); C 21:0 (heneicosanoic acid); C 20:1 (eicocyanic acid); and C 10:0 (capric acid).

Similar to the amino acids, 10 fatty acids had a higher value in the shrimp head, such as C 18:2 W6C (C-linoleic acid); C 18:1 W9C (C-oleic acid); C 17:1 (heptadecanoic acid); C 16:1 (palmitoleic acid); C 16:0 (palmitic acid); omega 9 fatty acids; C 12:0 (lauraic acid); monounsaturated fat; C 20:0 (arachidic acid); and saturated fat.

## 4. Discussion

After the shrimp were maintained in the pond for 40 days, the tiger shrimp juveniles fed with dietary phytoimmune-extract product made from *B. pandurata* and *S. ferox* at a dose of 20 and 30 mL kg^−1^ showed a significant difference in value compared with the control treatment (P0) for the average body weight (g)/ABW, average daily growth/ADG, and SGR (%). The increased growth rate was caused by the application of non-isoenergic diet for shrimp juveniles. In addition, the increased growth rate of shrimp juveniles occurred as a result of the application of seaweed extract [8]. The plant extract could be utilized as a drug or a feed additive ingredient for fish and shrimp [26]. 

Shrimp fed with 30 mL kg^−1^ phytoimmune-extract-supplemented diet obtained the maximum average body weight (7.2 g) after 40 days of the maintenance period in the pond. This result was closed to the tiger shrimp growth rate after being fed with seaweed extract [26] and *Sargassum wightii* extract [27] at 7.09-8.54 g. Several plant extract applications, such as *Laurencia snyderiae*, *Hypnea cervicornis*, and *Crypto nemia,* also presented a positive effect on the growth of white shrimp *L. vannamei* [28,29]. The increased shrimp growth rate as mentioned above could be associated with the water quality of pond, vitamin and mineral contents, and the improvement in the diet nutrient absorption efficiency ratio in shrimp was found to be similar to that in [30]. The water quality of the traditional pond was in normal condition (Table 6). Its mean traditional management was preservation of the DO, pH, and salinity.

Crustacea such as shrimp have no adaptive or specific immune system, and thus only rely on the innate immunity (non-specific immune system), as an evolutionarily older immune strategy [31,32]. The most distinctive immune system in shrimp is a cellular immune system [31,33]. The granular and agranular cells in shrimp are responsive cells against pathogens and other stressors [27].

The cellular immune response of *P. monodon* involves hemocytes, induced by the activation process of phenol oxidase (PO) through the phagocytosis process [1]. In this study, the application of 30 mL kg^−1^ phytoimmune extract could increase the maximum phenol oxidase activity (PO) significantly (*p* < 0.05). In addition, an increased SO level occurred significantly in the 20 and 30 mg L^−1^ phytoimmune-extract treatment groups on the 30th day of the maintenance period (*p* < 0.05). These data reveal a positive effect of *B. pandurata* and *S. ferox* (phytoimmune extracts) for improving the non-specific immunity of tiger shrimp. The cellular immune response is the primary immune response in shrimp against pathogens and stress. Both the PO and SO levels indicate the shrimp’s health.

The dietary supplementation of the plant extract could increase the total hemocytes of the tiger shrimp [34], similar to the application of phytoimmune-extract dietary supplementation in this study, which gradually increased the total hemocytes. Another study that followed this result was found on the supplementation of red seaweed *Gracilaria fisheri* extract in *P. monodon*, which showed an increased THC value in the post-supplementation period [35]. In shrimp, hemocytes play an important role in cellular immune system, and are more sensitive to pathogens [36]. One of the effective immune responses in the invertebrates is the phenol oxidase system. When the phenol oxidase enzyme activity declines, the phagocytosis process will fail. The hemocytes activated will simultaneously produce other bactericidal substances, such as H_2_O_2_ and superoxide anion (O_2_), and help induce the resistance level to infectious and non-infectious diseases [37]. Shrimp, unlike fish, lack leukocytes and must rely on hemocytes to transport nutrients. Hemocytes in shrimp contain granular, agranular, and hyaline cells. Increased THC levels indicate that the shrimp are in good health and are growing well.

The total hemocytes of tiger shrimp gradually increased from the first 10 days after the phytoimmune-extract dietary supplementation treatments, as the highest total hemocyte value was found in the 30 mL kg^−1^ diet dose at 5.7 ± 0.2 × 10^5^ cells mL^−1^. The increased TH in shrimp also occurred in another dietary extract supplementation dose, which indicates that the cell performance improved well for producing TC compared with the control treatment without the dietary phytoimmune-extract supplementation.

Hemocytes have an important role in cellular immunity, although migrating continuously as in macrophages in fish, which have a high sensitivity level against pathogens [36]. The effective immune system in the invertebrates is a phenol oxidase system. When the phenol oxidase enzyme activity declines, the phagocytosis process will fail. As mentioned in this study, the increased TH was also followed by the increased PO and SO. 

PO will be formed when the ProPO reacts with several compounds, such as zymosans (carbohydrates of yeast cell wall), bacterial lipopolysaccharides (LPS), urea, calcium ion, and tripsin [26]. In this study, shrimp fed with the 20 and 30 mL kg^−1^ phytoimmune-extract-supplemented diet showed a significantly higher PO concentration than the control diet group. The immunology parameter revealed the gradual increase in PO activity on the 10th day in the P2 treatment and the 20th day in the P1 treatment, which indicates that the shrimp larvae immune system was improved. The increased TH, PO, and SO levels has also been shown in *P. monodon* supplemented with the β-glucan immunostimulatory herbs [8,27,37].

The increased immune system in shrimp fed with the phytoimmune-extract-supplemented diets provided a good protection during the maintenance period in the pond, as a low mortality level at only 10–15% occurred in the phytoimmune extract application treatment groups. The tiger shrimp growth performance also significantly increased on the 40-th day of the maintenance period, although showing no significant difference between the 20 and 30 mL kg^−1^ doses. Only ABW obtained a significant different value between the different doses.

Shrimp contain an adequate amount of these omega-3 fats, and the fatty acid composition, as well as the MUFA/SFA, PUFA/SFA, w-3/w-6, and EPA/DHA fatty acid ratios, differed between species and diet. The variation of shrimp culture method, shrimp feed, and water quality could be attributed to differences in shrimp amino acid and fatty acid composition. When comparing meat and poultry, we conclude that shrimp are one of the most nutritious foods. Thus, the current study emphasizes that the shrimp are nutritionally dense, containing all of the proteins, lipids, amino acids, and fatty acids, and can serve as an effective diet supplement, as well as be encouraged for aquaculture under controlled environmental conditions. 

The amounts of AA, EPA, and DHA in tiger prawns with the extract were higher than those without the extract, with high DHA of 0.3149%, EPA of 0.4199%, and AA of 0.5686%. The amount of DHA (C22:6w-3) was higher than the amount of EPA (C20:5w-3) based on research of Bragagnolo and Rodriguez-Amaya, some species of shrimp *Xiphopenaeus kroyeri* [38], and pink shrimp, *Parapenaeus longirostris* [39] were similar. According to Oksuz et al. [40], the DHA content of *P. monodon* and *P. vannamei* was higher than that of EPA. The content of AA and EPA was higher than the DHA in this study, indicating that the fatty acid composition of *P. monodon* shrimp muscle was influenced by feed, size, salinity, temperature, season, and the Mahakam Delta’s geographic location.

Traditional culture, using plant extracts, showed a satisfying yield. Besides shrimp growth and survival rate, the quality of meat containing amino acids and fatty acids was also increased. Shrimp farmers and local governments should reconsider using plant extract products in aquaculture because it provides more economic and environmental benefits. 

## 5. Conclusions

Many factors influence the shrimp production level in ponds, specifically in the tradition pond system management performed by the community in East Borneo. This study presents an illustration and description of the phytoimmune extract supplementation efficacy, containing hairy-fruited eggplant and ginger extract at 2:1 ratio, for improving the growth performance (ABW, ADG, and SGR); inducing the TH, phenol oxidase activity (PO), anion superoxide (SO), and suppressing the shrimp mortality level when cultured in a pond. Both diets for 20 and 30 mL kg^−1^ dose of phytoimmune extract obtained a similar efficacy level. However, specific studies to identify the specific compounds in hairy-fruited eggplant and ginger that have an active role both in tiger shrimp growth performance and immunity response should be conducted further.

## Figures and Tables

**Figure 1 pathogens-11-01243-f001:**
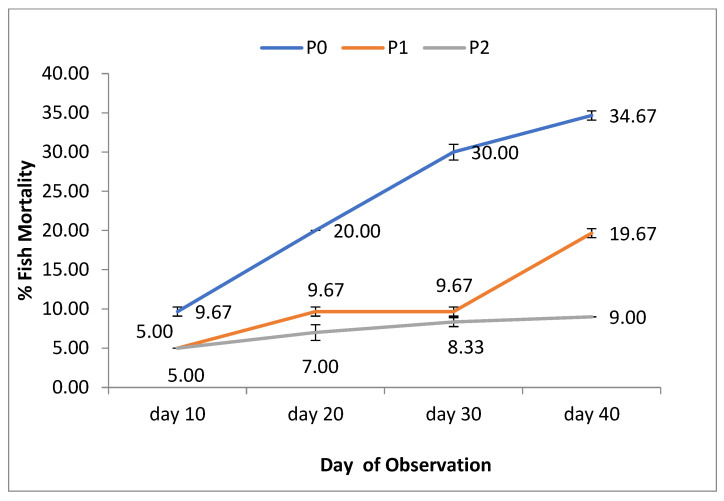
The *P. monodon* mortality percentage fed with phytoimmune supplemented diet and control diet for 40 days.

**Table 1 pathogens-11-01243-t001:** Tiger shrimp diet formulation (g kg^−1^) with phytoimmune extract supplementation.

Composition	P0	P1	P2
Shrimp head meal (g)	180	180	180
Fish meal (g)	320	320	320
Wheat flour (g)	220	220	220
Gluten meal (g)	60	60	60
Rice flour (g)	60	60	60
Soybean meal	100	100	100
Fish oil (mL)	20	20	20
Strach (g)	20	20	20
Mineral mix (g)	20	20	20
Phytoimmune ^1^ (mL)	0	20	30
Approximate feed composition			
Water content (%)	8.27	8.3	8.22
Ash (%)	15.18	16.13	17.42
Crude protein (%)	35.22	35.4	35.53
Crude lipid (%)	2.26	3.42	3.1
Carbohydrate (%)	39.07	36.75	35.73

^1^*B. pandurata* and *S. ferox*.

**Table 2 pathogens-11-01243-t002:** *P. monodon* growth performance after feeding with the diet treatment groups for 40 days.

Composition	Group	Observation Period			
		Day 10	Day 20	Day 30	Day 40
ABW (g)	P0	1.1 ± 0.1 ^a^	2.1 ± 0.4 ^b^	2.4 ± 0.1 ^b^	3.9 ± 0.3 ^c^
	P1	2.2 ± 0.2 ^b^	4.1 ± 0.1 ^c^	5.3 ± 0.0 ^c^	6.2 ± 0.1 ^d^
	P2	2.5 ± 0.2 ^b^	4.8 ± 0.2 ^c^	6.4 ± 0.2 ^d^	7.2 ± 0.2 ^e^
ADG (g day^−1^)	P0	0.11 ± 0.1 ^a^	0.11 ± 0.1 ^a^	0.08 ± 0.1 ^a^	0.10 ± 0.1 ^a^
	P1	0.22 ± 0.1 ^b^	0.21 ± 0.1 ^b^	0.18 ± 0.1 ^b^	0.16 ± 0.1 ^b^
	P2	0.25 ± 0.1 ^c^	0.24 ± 0.1 ^c^	0.21 ± 0.1 ^c^	0.18 ± 0.1 ^b^
SGR (%)	P0	0.95 ± 0.1 ^a^	3.71 ± 0.1 ^a^	2.92 ± 0.1 ^a^	3.40 ± 0.1 ^a^
	P1	7.88 ± 0.1 ^b^	7.05 ± 0.1 ^b^	5.56 ± 0.1 ^b^	4.56 ± 0.1 ^b^
	P2	9.16 ± 0.1 ^b^	7.84 ± 0.1 ^b^	6.19 ± 0.1 ^b^	4.94 ± 0.1 ^b^

Different superscript letters on the same line for each parameter show a significant difference (*p* < 0.05).

**Table 3 pathogens-11-01243-t003:** The immunomodulatory activity of shrimp fed with extract-supplemented diets and control diet. The total hemocytes (TH), phenol oxidase activity (PO), and anion superoxide concentration (SO).

Composition	Group	Observation Period			
		Day 10	Day 20	Day 30	Day 40
TH (10^5^ cells mL^−1^)	P0	4.9 ± 0.4 ^a^	4.7 ± 0.2 ^a^	5.1 ± 0.2 ^a^	5.1 ± 0.5 ^a^
	P1	5.0 ± 0.2 ^a^	5.1 ± 0.1 ^a^	5.4 ± 0.1 ^b^	5.4 ± 0.2 ^b^
	P2	5.2 ± 0.3 ^a^	5.5 ± 0.4 ^b^	5.6 ± 0.4 ^b^	5.7 ± 0.2 ^b^
PO (OD 490 nm)	P0	0.19 ± 0.04 ^a^	0.19 ± 0.04 ^a^	0.19 ± 0.06 ^a^	0.20 ± 0.02 ^a^
	P1	0.18 ± 0.01 ^a^	0.19 ± 0.04 ^a^	0.20 ± 0.07 ^a^	0.20 ± 0.01 ^a^
	P2	0.20 ± 0.02 ^b^	0.20 ± 0.02 ^b^	0.21 ± 0.04 ^b^	0.23 ± 0.03 ^b^
SO (OD 630 nm)	P0	0.05 ± 0.01 ^a^	0.06 ± 0.01 ^a^	0.06 ± 0.02 ^a^	0.06 ± 0.01 ^a^
	P1	0.06 ± 0.01 ^a^	0.07 ± 0.00 ^a^	0.09 ± 0.01 ^b^	0.08 ± 0.00 ^b^
	P2	0.07 ± 0.00 ^a^	0.07 ± 0.01 ^a^	0.09 ± 0.01 ^b^	0.09 ± 0.01 ^b^

Different superscript letters on the same line for each parameter show a significant difference (*p* < 0.05).

**Table 4 pathogens-11-01243-t004:** Several amino acid contents (mg kg^−1^) in shrimp meat and heads after feeding with 30 mL kg^−1^ of phytoimmune extracts and control diets.

Amino Acid Type (mg kg^−1^)	Shrimp Meat		Shrimp Head	
	Control	P2	Control	P2
L-Cystine	56,413.07	56,466.18	24,160.76	24,180.44
L-Methionine	8500.58	8520.86	2916.78	2918.05
L-Serine	24,633.41	24,716.14	14,953.73	14,999.02
L-Glutamic Acid	90,620.8	91,109.07	34,657.86	34,566.42
L-Phenylalanine	25,928.48	25,979.99	14,079.15	14,086.56
L-Isoleucine	24,248.84	24,341.69	11,318.36	11,345.83
L-Valine	25,367.82	25,549.14	14,749.06	14,810.70
L-Alanine	34,077.31	34,275.23	17,059.32	17,115.16
L-Arginine	52,698.91	52,975.34	16,568.30	16,625.57
Glycine	44,871.87	45,070.06	20,865.24	20,931.82
L-Lysine	51,634.89	51,881.43	16,421.97	16,475.75
L-Aspartic Acid	52,336.6	52,583.77	23,428.32	23,458.00
L-Leucine	43,825.49	44,013.44	17,229.53	17,268.10
L-Tyrosine	19,491.9	19,611.29	8731.65	8746.47
L-Proline	16,979.02	17,039.32	12,610.90	12,623.35
L-Threonine	27,163.5	27,301.99	14,497.80	14,522.81
L-Histidine	12,748.99	12,754.34	7409.95	7458.42
L-Tryptophan	2867.86	2873.6	2781.41	2795.45
Taurine	1349.4	1372.01	5609.91	5626.02

**Table 5 pathogens-11-01243-t005:** Several fatty acid contents (%) in shrimp meat and head after feeding with 30 mL kg^−1^ of phytoimmune extracts and control diets.

Amino Acid Type (^%^)	Shrimp Meat		Shrimp Head	
	Control	P2	Control	P2
Linolenic Acid	0.0435	0.0452	0.0267	0.0272
Linoleic Acid	0.2582	0.2593	0.2632	0.2714
Oleic Acid	0.4290	0.4312	1.0678	1.0756
C 18:2 W6 (Linoleic Acid/W6)	0.2582	0.2593	0.2632	0.2714
C 18:2 W6C (C-Linoleic Acid)	0.2582	0.2593	0.2632	0.2714
C 18:1 W9C (C-Oleic Acid)	0.4290	0.4312	1.0678	1.0756
C 20:5 W3 (Eicosatpentaenoic Acid)	0.4188	0.4199	0.1193	0.1227
C 17:1 (Heptadecanoic Acid)	0.0365	0.0378	0.0454	0.0477
C 16:1 (Palmitoleic Acid)	0.0758	0.0788	0.0925	0.0947
C 20:4 W6 (Arachidonic Acid)	0.5616	0.5686	0.3444	0.3457
Omega 6 Fatty Acids	0.6234	0.8198	0.6089	0.6158
C 20:2 (Eicosadienoic Acid)	0.0290	0.0305	0.0268	0.0276
DHA	0.3070	0.3149	0.1243	0.1256
Omega 3 Fatty Acids	0.7721	0.7771	0.2720	0.2736
C 18:3 W3 (Linolenic Acid/W3)	0.0435	0.0452	0.0267	0.0272
C 24:0 (Lignoseric Acid)	0.0831	0.0849	0.0534	0.0543
Polyunsaturated Fat	1.7888	1.7917	1.0752	1.076
C 22:6 W3 (Docosahexanoic Acid)	0.307	0.3149	0.1243	0.1256
C 18:0 (Stearic Acid)	0.5055	0.5084	0.4736	0.4806
C 22:2 (Docosadinoic Acid)	0.1612	0.1628	0.1603	0.165
C 17:0 (Heptadecanoic Acid)	0.0968	0.1000	0.0702	0.0721
C 16:0 (Palmitic Acid)	0.5478	0.555	0.9938	1.0023
Unsaturated Fat	2.3324	2.3373	2.2832	2.2907
Omega 9 Fatty Acids	0.429	0.4312	1.0678	1.0756
C 15:0 (Pentadecanoic Acid)	0.0243	0.0251	0.0171	0.0171
AA	0.5616	0.5686	0.3444	0.3577
C 14:0 (Myristic Acid)	0.0244	0.025	0.036	0.038
EPA	0.4188	0.4199	0.1193	0.1227
C 12:0 (Lauraic Acid)	0.0099	0.0102	0.0337	0.0348
Monounsaturated Fat	0.5436	0.5456	1.208	1.2157
C 20:0 (Arachidic Acid)	0.00145	0.00156	0.0237	0.0241
Saturated Fat	1.2976	1.3027	1.7068	1.7193

**Table 6 pathogens-11-01243-t006:** Average pond water quality.

Parameter	Unit	Value
Temperature	°C	18–30
Salinity	‰	15–25
pH		5.5–8
Dissolved oxygen (DO)	mg L^−1^	4–5
pH of pond substrate	-	6–8
Pyrite	‰	1.46–2.98

## Data Availability

Not applicable.

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
