# Peer review of "Application of Phyto-Stimulants for Growth, Survival Rate, and Meat Quality Improvement of Tiger Shrimp (Penaeus monodon) Maintained in a Traditional Pond"

_pathogens, 2022, doi:10.3390/pathogens11111243_

Round 1
Reviewer 1 Report
Notes are integrated in MS pdf

Author Response
Dear Reviewer 1,
Thank you for reviewing a revised draft of the manuscript “Application of phyto-stimulant as a natural fish drug for growth, survival rate, and meat quality improvement of tiger shrimp maintained in a silvofishery pond ", submitted to pathogens. We appreciate the time and effort that you and the reviewers dedicated to providing feedback on our manuscript and are grateful for the insightful comments on and valuable improvements to our paper. We have incorporated most of the suggestions made by the reviewers. Those changes are highlighted in the manuscript. Please see the attachment, for a point-by-point response to the reviewer's comments and concerns.

Reviewer 2 Report
The work on “Application of phytoimmune as a natural fish drug for growth, survival rate, and meat quality improvement of tiger shrimp maintained in a silvofishery pond” carried out by Hardy et al. is well executed and an area of interest in the present scenario. There have been several reported works on phytomedicines, however, the practical relevance of the study is reflected. Before recommending for publication of the manuscript, I have suggested few changes to be made. The paper needs rewriting. My section-wise observations and comments are highlighted below for the authors to work on.
I recommend the publication of the article upon a major revision.
Title
The word phytoimmune seems confusing. I believe there are alternative words, like biomedicine, phyto-stimulants etc. Using the word drug is irrelevant for plant-based extracts.
Abstract
Line 23: use of word performed seem very untechnical. May replace with cultured.
Line 24-26: Sentence needs rewriting.
In fact, the abstract should give a simple message of the entire content. The way it is presented in its current form is unacceptable.
Also, there are spelling mistakes in a few keywords.
Introduction
The introduction needs rewriting. The English language is poor. Give special attention to the challenges of shrimp farming in East Borneo, followed by a few comparisons with standard farming practices in other countries. This should be followed by why herbal medicines are preferred, referring to a few studies on the species. Detail more about the products that are being examined, and explain why it is so important in your context. Last, briefly describe the various parameters considered in your study, followed by the objectives of your study. Avoid using words like “article” because yours is a study.
Materials and methods
1. Line 78: Tiger shrimp can be replaced with “experimental animal”
2. Line 96-97: Confusing sentence. Please explain.
3. Line 108: Feeding activity can be simply written as feeding.
4. Line 113: This section can be subdivided.
5. Line 143-157: Please delete these lines.
Results
1. Use probability sign as p or P. Should be uniform throughout.
2. Line 167: In heading its survival %. Why describe the mortality rate here.
3. mgL-1 should be changed to mgL-1 in all cases
4. Table 1: diet formulation should explain the sources of the ingredients.
5. Table 2. All significant values should be stated in superscripts
6. SI Units should be the same in all cases. Many mistakes are observed.
Discussion
This section forms the most crucial part of the MS. Elaborate more and discusses the present finding with proper reasoning. I suggest further improvement.
The conclusion needs to be very concrete with a strong takeaway message.
Author Response
Dear Reviewer 2,
Thank you for advising our manuscript. We appreciate the time and effort that you and the reviewers dedicated to providing feedback on our manuscript and are grateful for the insightful comments on and valuable improvements to our paper. We have incorporated most of the suggestions. Those changes are highlighted in the manuscript. Please see below, for point-by-point responses.
Point 1: Title.
The word phytoimmune seems confusing. I believe there are alternative words, like biomedicine, phyto-stimulants etc. Using the word drug is irrelevant for plant-based extracts.
Response 1: Application of phyto-stimulants for growth, survival rate, and meat quality improvement of tiger shrimp (Penaeus monodon) maintained in a silvofishery pond
Point 2: Abstract.
Line 23: use of word performed seem very untechnical. May replace with cultured.
Line 24-26: Sentence needs rewriting.
In fact, the abstract should give a simple message of the entire content. The way it is presented in its current form is unacceptable.
Also, there are spelling mistakes in a few keywords.
Response 2: thank you for your advice. I have replaced the word.
Point 3: Introduction.
The introduction needs rewriting. The English language is poor. Give special attention to the challenges of shrimp farming in East Borneo, followed by a few comparisons with standard farming practices in other countries. This should be followed by why herbal medicines are preferred, referring to a few studies on the species. Detail more about the products that are being examined, and explain why it is so important in your context. Last, briefly describe the various parameters considered in your study, followed by the objectives of your study. Avoid using words like “article” because yours is a study.
Response 3: I Rewrite the introduction.
Point 4: Materials and methods.
- Line 78: Tiger shrimp can be replaced with “experimental animal”
- Line 96-97: Confusing sentence. Please explain.
- Line 108: Feeding activity can be simply written as feeding.
- Line 113: This section can be subdivided.
- Line 143-157: Please delete these lines.
Response 4: Thank you, I have checked and corrected your suggestion.
Point 5: Results
- Use probability sign as p or P. Should be uniform throughout.
- Line 167: In heading its survival %. Why describe the mortality rate here.
- mgL-1 should be changed to mgL-1 in all cases.
- Table 1: diet formulation should explain the sources of the ingredients.
- Table 2. All significant values should be stated in superscripts.
- SI Units should be the same in all cases. Many mistakes are observed.
Response 5:
Thank you for the revision, I checked and replaced it with the right form.
Point 6: Discussion.
This section forms the most crucial part of the MS. Elaborate more and discusses the present finding with proper reasoning. I suggest further improvement.
Response 6:
Note.
Point 7: The conclusion needs to be very concrete with a strong takeaway message.
Response 7: Thank you, I have checked and deleted it because of part of the manuscript’s template.
Reviewer 3 Report
In this manuscript, the authors examined the growth rate, mortality, associated enzyme activity, and amino and fatty acid content of tiger shrimp after adding plant extracts. However, this manuscript has a lot of mistakes, and the amount of data provided is somewhat limited. There is no sufficient discussion of the results obtained. The general impression of the submitted paper is that it has been written instead in a hurry with poor attention to the number of important details.
Some comments are added below that could help authors to improve this manuscript exhaustively.
1. In the Abstract, there are many descriptions of methods in this part. Still, the research purpose is unclear, the summary of results is incomplete, and important enzyme activity results are ignored. It is recommended to rewrite.
2. L40, add the Latin name of tiger shrimp.
3. L99,100, revise “mg L-1” to “mg·L-1”, Please check and change the writing of unit symbols throughout the text.
4. L143-157, Why are these paragraphs in the article? I hope the author will be more careful and patient in the writing process.
5. L186, revise “p” to “P”.
6. L243-244, This sentence is not clearly stated whether the growth rate increased or decreased after feeding plant extracts compared with the control group. The authors should write the results.
7. Figure 1, There is no standard deviation in the graph. Did the authors perform the statistical analysis?
8. Table 1-4, The position of the first row of units is mislabeled. For example, in Table 1, (g) should be after P0, P1, and P2.
9. No discussion of fatty acid and amino acid content was seen in the Discussion, and the discussion was not deep enough.
Author Response
Dear Reviewer 3,
Thank you for your suggestions. We appreciate the time and effort that you and the reviewers dedicated to providing feedback on our manuscript and are grateful for the insightful comments on and valuable improvements to our paper. We have incorporated most of the suggestions. Those changes are highlighted in the manuscript. Please see below, for a point-by-point response to the reviewer comments and concerns.
Point 1: In the Abstract, there are many descriptions of methods in this part. Still, the research purpose is unclear, the summary of results is incomplete, and important enzyme activity results are ignored. It is recommended to rewrite.
Response 1: Thank you I added some information in the abstract.
Point 2: L40, add the Latin name of tiger shrimp.
Response 2: Done.
Point 3: L99,100, revise “mg L-1” to “mg·L-1”, Please check and change the writing of unit symbols throughout the text.
Response 3: Done.
Point 4: L143-157, Why are these paragraphs in the article? I hope the author will be more careful and patient in the writing process.
Response 4: Done, thank you very much.
Point 5: L186, revise “p” to “P”.
Response 5: Done
Point 6: L243-244, This sentence is not clearly stated whether the growth rate increased or decreased after feeding plant extracts compared with the control group. The authors should write the results.
Response 6: Done, with adding some reasons and explanation.
Point 7: Figure 1, There is no standard deviation in the graph. Did the authors perform the statistical analysis?
Response 7: I Replaced the graphic.
Point 8: Table 1-4, The position of the first row of units is mislabeled. For example, in Table 1, (g) should be after P0, P1, and P2.
Response 8: Replace
Point 9: No discussion of fatty acid and amino acid content was seen in the Discussion, and the discussion was not deep enough.
Response 9: Adding the information about the meat quality of shrimp.
Round 2
Reviewer 1 Report
Thank you for your corrections.
Author Response
Thank you very much for your kindly review my article.
Reviewer 2 Report
The authors have carefully addressed the suggestions. No more comments.
Author Response
thank you for taking the time to review my article. your suggestion makes the article better.
Reviewer 3 Report
I have carefully reviewed the Full-Length Article entitled “Application of phyto-stimulants for growth, survival rate, and meat quality improvement of tiger shrimp (Penaeus monodon) maintained in a traditional pond”. Authors attended most of the suggestions and the manuscript has improved substantially. I only have a minor issue with "the authors need to adjust the order of references in the article".
Author Response
Thanks for the cerefully advice, I have corrected the reference section by following GFA from Pathogens Journal.